# Discrepancies between Promised and Actual AI Capabilities in the Continuous Vital Sign Monitoring of In-Hospital Patients: A Review of the Current Evidence

**DOI:** 10.3390/s24196497

**Published:** 2024-10-09

**Authors:** Nikolaj Aagaard, Eske K. Aasvang, Christian S. Meyhoff

**Affiliations:** 1Department of Anaesthesia and Intensive Care, Copenhagen University Hospital—Bispebjerg and Frederiksberg, 2400 Copenhagen, Denmark; christian.sylvest.meyhoff@regionh.dk; 2Department of Anaesthesia, Centre for Cancer and Organ Diseases, Copenhagen University Hospital—Rigshospitalet, 2100 Copenhagen, Denmark; eske.kvanner.aasvang.01@regionh.dk; 3Department of Clinical Medicine, University of Copenhagen, 2200 Copenhagen, Denmark

**Keywords:** continuous vital sign monitoring, wireless biosensors, artificial intelligence, machine learning, acute complications, serious adverse events, adverse events, medical patients, surgical patients

## Abstract

Continuous vital sign monitoring (CVSM) with wireless sensors in general hospital wards can enhance patient care. An artificial intelligence (AI) layer is crucial to allow sensor data to be managed by clinical staff without over alerting from the sensors. With the aim of summarizing peer-reviewed evidence for AI support in CVSM sensors, we searched PubMed and Embase for studies on adult patients monitored with CVSM sensors in general wards. Peer-reviewed evidence and white papers on the official websites of CVSM solutions were also included. AI classification was based on standard definitions of simple AI, as systems with no memory or learning capabilities, and advanced AI, as systems with the ability to learn from past data to make decisions. Only studies evaluating CVSM algorithms for improving or predicting clinical outcomes (e.g., adverse events, intensive care unit admission, mortality) or optimizing alarm thresholds were included. We assessed the promised level of AI for each CVSM solution based on statements from the official product websites. In total, 467 studies were assessed; 113 were retrieved for full-text review, and 26 studies on four different CVSM solutions were included. Advanced AI levels were indicated on the websites of all four CVSM solutions. Five studies assessed algorithms with potential for applications as advanced AI algorithms in two of the CVSM solutions (50%), while 21 studies assessed algorithms with potential as simple AI in all four CVSM solutions (100%). Evidence on algorithms for advanced AI in CVSM is limited, revealing a discrepancy between promised AI levels and current algorithm capabilities.

## 1. Introduction

Delayed detection of patient deterioration in hospital wards significantly contributes to in-hospital mortality and intensive care unit (ICU) admissions [1,2]. As a consequence, routine manual vital sign monitoring at fixed intervals, such as the National Early Warning Score (NEWS) has been implemented across countries [3,4]. However, no documented impact of NEWS on morbidity and mortality has been observed [5,6]. This may be partly due to deterioration occurring between manual observations, where continuous vital sign monitoring (CVSM) with wireless sensors can enable the early detection of patient deterioration, which is important for timely treatment and prevention of further decline [7,8,9]. Clinical implementation of CVSM may reduce the risk of serious adverse events (SAEs) [10,11], as it has been documented to detect more vital sign deviations [12,13]. NEWS-based algorithms have demonstrated an inconsistent ability to predict subsequent SAEs when applied to CVSM [14,15]. Identifying the optimal thresholds for various SAE subtypes is thought to be crucial and potentially attainable through advanced data analysis and artificial intelligence (AI) [16]. Thus, AI implementation to sensor data may enhance sensitivity and specificity, ensuring that CVSM sensors are effective, and assist clinical staff without causing alarm fatigue or unintended adverse events (AE) from overtreatment and unnecessary diagnostic procedures [16]. AI encompasses a wide range of technologies designed to simulate or replicate human cognitive functions, like problem solving and decision making, and the foundation of AI algorithms varies widely in terms of data and analysis methods [17]. Despite significant advancements evidenced in publications, the implementation of these technologies in the medical field remains slow and complex [18,19] and a discrepancy between what is promised on CVSM producers websites and what is documented in the scientific literature may therefore exist. It is crucial to validate and report the clinical impact of AI algorithms used for CVSM sensors in peer-reviewed studies to ensure transparency and enable decision makers to determine when and what to implement [16]. We hypothesized that CVSM producers are promising AI advancements beyond what has been evaluated in peer-reviewed evidence. This review aims to summarize current research on CVSM sensors in hospitalized patients with potential for application as AI algorithms within state-of-the-art CVSM solutions.

## 2. Materials and Methods

### 2.1. Litterature Search

We conducted an electronic literature search in PubMed and Embase (Ovid). We searched for articles published up to 30 May 2024. No limits were set for publication date in PubMed, while studies were limited to those published after 2010 in the Embase search. The search was conducted using Medical Subject Headings in PubMed. No limitations were applied to the search in PubMed. In Embase, the following limits were applied: human, English language, study type (clinical trial, randomized controlled trial, controlled clinical trial, or multicenter study), adult (age 18–64 years), and elderly (age > 64 years). Based on title and abstract, the search results were screened and assessed for eligibility with Rayyan (http://rayyan.qcri.org) [20]. The complete search strategy, including terms, is detailed in Table 1. The official websites of the CVSM solutions, identified through the literature search, were also examined for peer-reviewed evidence and other relevant material, including white papers and reports [21,22,23,24,25]. If a CVSM study was excluded at full-text review, the official website of the solution was still evaluated, and any relevant evidence was retrieved and assessed accordingly. Peer-reviewed evidence from these websites was retrieved via PubMed or Embase and assessed. Non-peer-reviewed evidence was retrieved and evaluated in the same way as peer-reviewed evidence.

### 2.2. Study Selection

The inclusion criteria in this review were based on the following PICO elements:

#### 2.2.1. Population

We included studies on adult patients (age ≥ 18 years), admitted to general medical or surgical wards. Studies focusing on ICUs, post-anesthesia care units, and other high-dependency units were excluded, as CVSM is already standard in these settings, and the potential AI algorithms would likely be different due to a higher staff–patient ratio in the handling of alerts. Studies on non-hospitalized patients were also excluded, as these patients are typically not monitored at all. Studies on invasive CVSM sensors were excluded as they are not typically practical or standard in general wards.

#### 2.2.2. Intervention

CVSM was defined as non-invasive recordings of one or more vital signs continuously at a high sampling frequency (e.g., less than five minutes) with one or more wireless biosensors. This review included studies measuring traditional vital signs, including respiratory rate (RR), peripheral oxygen saturation (SpO_2_), heart rate (HR), temperature (Tp), and/or blood pressure (BP), with BP considered semi-continuous with intermittent measurements of less than an hour with the same wireless device applied throughout the monitoring period. Only sensors intended for medical use were included, i.e., studies on consumer-grade fitness trackers were excluded. Only studies of CVSM solutions claiming on their official websites to be supported by AI or algorithms that generate alarms for patient deterioration or complications were included.

#### 2.2.3. Outcome

The outcome of interest was the classification of AI level as reported on product websites and reported in peer-reviewed articles. Several definitions and classifications of AI exist, and for this study, AI classification was based on evaluation of the statistical methods or algorithms used to predict complications or set alarms within the CVSM solution, with the level of AI determined by its ability to replicate human capabilities [26]. Simple AI (level 1) corresponds to reactive machines that have no memory or learning capabilities, thus restricting these systems or algorithms to automated responses for a limited number of parameters, specifically when crossing predefined or fixed vital sign thresholds [26]. Fixed vital sign thresholds that were adjusted during the study based on clinical staff feedback were also considered simple AI. Only studies evaluating thresholds for improving clinical outcomes, predicting complications or specific outcomes (e.g., ICU admission, mortality, readmission), and optimizing alarm thresholds were included. Studies quantifying the ability of CVSM sensors to detect deviations compared to routine monitoring or describing deviations from fixed thresholds without association to clinical outcomes were excluded, as they do not represent predictive algorithms with AI potential. If the study analyzed and described vital sign thresholds or patterns in specific cases prior to a complication, such as sepsis or myocardial infarction, it was graded as simple AI, provided that no adaptive algorithms were used for analysis.

Advanced AI (level 2) corresponds to limited memory models that can learn from past data to make decisions, thus they have adaptive capabilities and the ability to modify and adjust thresholds based on previous vital sign measurements. Studies utilizing machine learning (ML) methods, a subset of AI focused on developing algorithms and statistical models that learn from complex patterns and make data-based predictions [27], were classified as advanced AI. Only studies evaluating ML algorithms used to predict complications or set alarms were classified as advanced AI. ML algorithms used to improve the accuracy of vital sign measurements were not included as advanced AI. As the current most-advanced AI applications are encompassed by limited memory models, this level is considered advanced AI, and no levels beyond this were classified. The different levels of AI are illustrated in Figure 1.

Studies with the following designs were included: randomized controlled trials (RCT), cluster RCTs, and observational studies. Non-peer-reviewed materials, such as white papers, reports, conference articles, and abstracts were also included. All studies on CVSM in adult patients admitted to general hospital wards were retrieved for full-text review, as relevant outcomes may not have been reported in the abstracts.

### 2.3. Data Extraction

We extracted characteristics of each monitoring solution (country of origin, devices used, regulatory status (CE/FDA), study design, type of included patients, and vital signs measured) and the promised AI capability level reported on the official websites. The method of analysis in studies assessing the prediction of complications or filtering of alarms was extracted and the potential AI level was evaluated and graded. The performance metrics of the algorithm were also retrieved and reported, including quantification of predictive ability, measures for improving clinical outcomes, and the time spent outside thresholds for patients with complications compared to those without.

## 3. Results

In total, 391 studies were identified through a search of the databases. Including peer-reviewed evidence from official websites, a total of 467 studies were considered, with 113 studies retrieved for full-text review, including nine different CVSM solutions. In total, 26 peer-reviewed studies of four different CVSM solutions supported by AI (ViSi Mobile^®^ (VM), WARD 24/7, Sensium Vitals^®^ (SV), and Biobeat) met the inclusion criteria (Figure 2).

The characteristics of each monitoring solution are presented in Table 2. Evidence from the official website of Isansys Lifecare was not evaluated independently, as it had been investigated in adult patients admitted to general wards through its integration with WARD 24/7 [28]. Four CVSM solutions identified through the literature search were not evaluated, as they do not currently claim AI or predictive algorithm support, or no eligible studies were identified [29,30,31,32,33]. The promised levels of AI compared to the highest potential AI levels of algorithms assessed in peer-reviewed studies are presented in Table 3.

A total of five studies assessing algorithms with the potential for advanced AI application have been published on two CVSM solutions (50%): four from WARD 24/7 [34,35,36,37] and one from SV [38]. A total of 21 studies assessing algorithms with the potential for simple AI applications have been published on all CVSM solutions (100%): six from VM [11,39,40,41,42,43], seven from WARD 24/7 [13,14,15,44,45,46,47], six from SV [48,49,50,51,52,53], and two from Biobeat [54,55] (Figure 3). The methods and reported outcomes of the included studies varied too much for a meta-analysis. Specifically, performance metrics were reported differently across studies, and even within subgroups of studies using similar metrics, the algorithms significantly differed and were modified during the studies. Therefore, the findings were presented descriptively. The following sections present the evidence for each solution.

### 3.1. Visi Mobile ^®^

The VM Surveillance Monitoring System (Sotera Digital Health, Carlsbad, CA, USA) is a CE-marked and FDA-cleared wireless wrist-worn device for monitoring vital signs, utilizing ML algorithms designed to identify patient deterioration and promptly alert hospital staff, while minimizing irrelevant alarms [21]. VM sends information directly to the medical records and the hospital staff’s phones, when connected to the hospital’s network (Table 2 and Table 3) [21].

#### 3.1.1. Improving Outcomes

A recent observational study monitored adult patients admitted to general surgical wards at a tertiary health care center (Atrium Health Wake Forest Baptist Medical Center) over a 2-year period (1 January 2018–31 December 2019). A propensity-matched analysis was performed and found that intermittent monitoring (n = 12,345) was associated with increased risk of mortality and ICU admission (odds ratio (OR) 3.42, 95% confidence interval (CI): 3.19–3.67), heart failure (OR 1.48, 95%CI: 1.21–1.81), myocardial infarction (OR 3.87, 95%CI: 2.71–5.71), and acute kidney injury (OR 1.32, 95%CI: 1.09–1.57) compared with CVSM (n = 7955) [11]. The odds of rapid response team (RRT) activation were similar in both groups (OR 0.86, 95%CI: 0.79–1.06) [11].

A recent before-and-after study of both surgical and medical patients from a university hospital in the Netherlands included 2303 and 2466 patients in the intervention and baseline cohort, respectively. Patients monitored with the VM system had significantly fewer unplanned ICU admissions with 3.4% vs. 2.3% (*p* = 0.03) and fewer RRT calls with 4.3% vs. 3.1% (*p* = 0.02). No differences in hospital length of stay or in-hospital mortality were observed [43]. Another before-and-after study found that the VM system, combined with modified early warning score, significantly reduced RRT activations from 9.6% in the baseline period (February to April 2016) to 4.8% in the intervention period (February to April 2018) (*p* < 0.01) [42]. An additional before-and-after study conducted from December 2015 to March 2016, in the orthopedic and trauma general care ward, found a significant reduction in complication incident rates of 34.3 per 1000 patient days and 9.6 per 1000 patient days for the preintervention and intervention group, respectively (*p* < 0.05) [41]. The frequencies of RRT calls were similar in the preintervention and intervention groups, while ICU transfers and failure to rescue were insignificantly reduced from 22 (5.4%) to 19 (4.5%) and 2 (0.46%) to 0, respectively [41].

#### 3.1.2. Alarm Filtering

A prospective, observational pilot study in neurological and neurosurgical patients evaluated the tolerability of alarm thresholds based on clinical staff feedback on 736 patients included for analysis [39]. Predefined vital sign thresholds were filtered for artifacts, and the alarm thresholds were adjusted over four iterations, aiming for two alarms per patient per day. An acceptable rate of 2.3 alarms per patient per day was achieved in the final stage of implementation with modified thresholds and durations (Table 4). The study reported maintaining patient safety with fewer RRT calls, ICU transfers, and unplanned deaths compared to the five months before implementation of the VM system, with the latter two not being statistically significant [39]. A different study aimed to reduce alarm fatigue and optimize alarm settings in CVSM using data from the VM system, analyzing 94,575 h from 3430 patients stored in cloud databases. By simulating various alarm thresholds and delays with cloud computing tools based on data distribution, the authors manually assessed alarm rates to potentially help manage alarm burden [40]. A white paper described alarm development within the VM system using the cloud database. Client-specific alarm data are processed through various “what-if” configurations, resulting in an alarm performance table that includes parameters, delays, vital sign thresholds, and corresponding alarms per patient per day [56]. An example of this alarm modification form is presented in Appendix A.

### 3.2. WARD 24/7

Wireless Assessment of Respiratory and circulatory Distress (WARD) (WARD 247 ApS, Copenhagen, Denmark) is a CE-marked clinical support system (CSS) with continuous 24 h vital sign monitoring of high-risk medical and surgical patients using advanced clinically modeled algorithms to monitor patients with real-time alarms [22,23]. Vital sign data are collected with the clinically validated CE- marked and FDA-approved Isansys Lifetouch and Lifetemp (Isansys Lifecare, Oxfordshire, UK), Nonin WristOx 3150 (Nonin Medical Inc., Plymouth, MN, USA), Meditech BlueBP-05 (Meditech Ltd., Budapest, Hungary), and A&D TM-2441 (A&D COMPANY, Tokyo, Japan) (Table 2 and Table 3) [22,23].

#### 3.2.1. Prediction of Serious Adverse Events and Deterioration

Studies with the WARD CSS have assessed the predictive ability of traditional vital signs for subsequent SAEs, defined according to the International Conference on Harmonization–Good Clinical Practice Guideline [57]. Two recent prospective observational trials, utilizing the WARD CSS assessed the predictive ability of predefined vital sign threshold, based on the NEWS, for subsequent SAEs in surgical and medical patients, respectively [14,15]. Both studies analyzed the duration and frequency of vital sign deviations and investigated the association between 24 h exposure periods and SAEs. Apart from bradypnea in the medical patients, both studies found no statistically significant association between the frequency or total duration of vital sign abnormalities and subsequent occurrence of SAEs [14,15]. For specific types of SAEs, one study in high-risk surgical and medical patients found a significant association between the accumulated duration of SpO_2_ < 85% and subsequent myocardial injury, with a 14.2 min longer deviation in those with upcoming myocardial injury compared to those without (95% CI: −4.7 to 33.1 min, *p* = 0.005) [44]. Other thresholds for SpO_2_ (<88% and <80%), HR (>110 and >130 beats/minute), and RR (>24 and >30 per minute) were also significantly associated with myocardial injury [44]. Another study in patients undergoing major gastrointestinal surgery found a higher detection rate of post-operative atrial fibrillation (6.5%, 95% CI: 4.5–9.4) based on CVSM compared to clinical staff observations (3.5%; 95% CI: 1.9–5.8) [47]. No significant association between preceding vital sign deviations and postoperative atrial fibrillation was identified [47]. A prospective observational study analyzed data from 90 patients with suspected COVID-19 infection. The study found that patients who died within 30 days or were admitted to the ICU had significantly longer durations outside the thresholds: RR > 21, RR > 24, SpO_2_ < 80%, SpO_2_ < 85%, and HR > 110 bpm [13]. A study examined the association between deviating vital signs prior to discharge and the risk of readmission. SpO_2_ < 88% for at least 10 min was observed in 66% of readmitted patients and 62% of non-readmitted patients (*p* = 0.62), while SpO_2_ < 85% occurred in 58% of readmitted and 52% of non-readmitted patients (*p* = 0.50) for at least 5 min. The sustained deviation of at least one vital sign was detected in 90% of readmitted and 85% of non-readmitted patients (*p* = 0.20). Deviating vital signs prior to discharge were frequent but not associated with increased readmission risk within 30 days [45].

#### 3.2.2. Alarm Filtering

A recent study evaluated the effect of alarm filtering on data from 716 medical and surgical patients included from four prospective observational trials. Various NEWS thresholds were applied to trigger alarms, which were then filtered by artifact removal combined with duration-specific criteria and artifact removal only. This significantly reduced alarms per patient per day from a median of 137 [IQR 87–188] to 101 [IQR 56–160] with only artifact removal, and 19 [IQR 9–34] when combining artifact removal with a duration-specific criterium (Table 4) [46].

#### 3.2.3. Future Possibilities

A recent study used Circadian Kernel density estimation to measure physiological stability [34]. This ML model was used to compare the final 24 h of monitoring for patients discharged from the hospital without experiencing any SAEs against the eight hours of data leading up to physiological deterioration, defined as occurrence of SAEs or total NEWS of either  ≥ 6, ≥ 8, or  ≥ 10 [34]. The model did not demonstrate a convincing ability to discriminate stable physiology from observations prior to SAEs with area-under-receiver-operating-characteristics curve (AUROC) values of 0.59–0.61, but effectively discriminated stable physiology from events of high NEWS with AUROC of 0.77–0.99, specifically with better discrimination for higher total NEWSs [34]. Another study evaluated four ML models for real-time SAE prediction [35]. The best performing model had an AUROC of 0.65 for patients with SAEs and a corresponding false-positive rate of 0.073 in patients without SAEs, as well as a balanced sensitivity and false-positive rate [35]. Three additional studies evaluated different ML models in a prospective observational study of patients undergoing major surgery. The first study included 453 patients and based on a support vector machine demonstrated high accuracy with an AUROC of 0.93 with a corresponding sensitivity of 80% and specificity of 93% for the prediction of any SAEs based on a 2 h prediction window and 10 h observation window [36]. Another study with 450 patients evaluated a SMOTE-enhanced XGBoost model for classifying nighttime vital signs as preceding an SAE or from patients without SAEs, achieving an AUROC of 0.65 [37].

### 3.3. Sensium Vitals ^®^

The SV patch (Sensium, Abingdon, UK) is a CE-marked and FDA-approved wireless CVSM solution, transmitting data to SV bridges in the ward [24]. These data can be integrated with electronic medical records, allowing for the input of additional patient data to calculate the NEWS [24]. SV smart algorithms continuously process and analyze patient data, generating targeted notifications of deterioration (Table 2 and Table 3) [24].

#### 3.3.1. Prediction of Serious Adverse Events

An observational study of 31 patients monitored with different CVSM solutions, including SV, descriptively analyzed vital sign trends in patients from a surgical step-down unit to general wards. Twenty AEs occurred in 11 patients, and the most common were atrial fibrillation and pneumonia, with four instances of each. SV effectively detected atrial fibrillation as a sudden HR increase and respiratory insufficiency as increased RF with decreased SpO_2_ [50]. No overall CVSM thresholds for AE prediction were highlighted, and specific vital sign changes for each case were not consistently detailed with thresholds [50]. A case-series based on implementation of SV in five European hospitals, including both medical and surgical wards, found that SV identified the first signs of respiratory and circulatory deterioration in the selected nine cases of AE with four cases of (paroxysmal) atrial fibrillation, two of sepsis, and one each of pyrexia, cardiogenic pulmonary edema, and pulmonary embolisms [53]. Specific thresholds were not consistently detailed but included descriptions of vital sign patterns. Tachycardia was noted in all cases and tachypnea in some for detecting atrial fibrillation [53]. A case study of a patient admitted for pneumonia and monitored with SV detected deterioration three hours post-admission due to a sudden increase in RR from 22/min to 51/min, HR from 75 to 157 bpm, and temperature from 38.0 to 39.5 °C, leading to a clinical examination and a new diagnosis of respiratory failure with severe sepsis [52].

#### 3.3.2. Improving Outcomes

A large RCT with 747 patients, of which 517 were in the control group and 230 in the intervention group, was terminated early (57% inclusion) due to COVID-19 [48]. Alarm limits were preset to HR < 40 or > 120, RR < 8 or >24, and Tp > 39.0 °C, and delayed for 14 min, before an audio alert was sent to nurses’ devices. New disability three months after surgery occurred in 43.7% of patients in the control group and 39.1% of patients in the intervention group, resulting in an absolute difference of 4.6%. Descriptive statistics showed that all groups of postoperative complications were detected equally or more often in the intervention group compared to the control group [48]. A feasibility RCT of surgical patients included 125 patients for analysis, with 60 patients in the intervention group. The control group experienced more complications across all Clavien–Dindo categories, notably in major complications with 15.5% (10 patients) in the control group versus 5.0% (3 patients) in the intervention group. The intervention group had fewer unplanned high-dependency unit and ICU admissions (5 vs. 1 patient), and a shorter average length of stay (11.6 days vs. 16.2 days) compared to the control group. The intervention group had higher readmission rates and longer delays in receiving antibiotics for sepsis cases [49]. Another pilot cluster RCT included 226 patients, with 140 randomized to CVSM with predefined thresholds. The study found that CVSM patients received antibiotics faster after sepsis (10.4 vs. 16.9 h), had shorter hospital stays (13.3 vs. 14.6 days), and had lower 30-day readmission rates (11.4% vs. 20.9%). None of the associations were statistically significant [51].

#### 3.3.3. Alarm Filtering

An observational study of 39 surgical patients simulated various alarm strategies, including SV threshold-based alarm algorithms and six methods for personalized alarm thresholds based on individual or situational factors (Table 4). A performance score, calculated from the overall alarm rate and early sensitivity defined as true-positive alarms within 24 h before an AE, evaluated alternative alarm strategies against the original strategy, with a positive value indicating improvement [38]. The SV algorithm produced an average of 0.49 alarms per patient per day, with an early sensitivity of 39% and a false detection rate of 59%. Compared to the original algorithm, only one strategy assessing lowered nighttime HR and RR thresholds, resulted in a positive performance score. A combination of strategies gave the best results, boosting early sensitivity up to 61% and overall sensitivity to 72%, with a slight increase in the alarm rate to 0.59 alarms per patient per day [38].

### 3.4. BioBeat

Biobeat (Biobeat Technologies Ltd., Petah Tikva, Israel) developed an FDA-cleared and CE-marked wrist monitor for repeated use and a disposable chest monitor for in-hospital monitoring, enabling the real-time monitoring of 14 cardiopulmonary vital signs processed every five minutes, including all standard vital signs as well as pulse pressure, systemic vascular resistance, heart rate variability, stroke volume, cardiac output, and cardiac index [25]. Biobeat states on their website that the system uses next-generation health AI and ML on big data to provide actionable insights for patient care, enabling a data-driven approach to personalized medicine (Table 2 and Table 3) [25].

#### Prediction of Deterioration

An observational study included 521 participants: 361 high-risk patients from general wards and 160 healthy controls. A multi-parameter real-time warning score (MPRT-WS) based on retrospectively defined thresholds set by a specialist panel for nine parameters, such as HR, RR, SpO_2_, Tp, and BP, was compared to the NEWS for detecting patient deterioration (cardiopulmonary resuscitation, ICU admission, death, or based on ABCDE criteria) (Appendix A) [55]. Comparing risk levels using both tools, NEWS classified 92.6% as low, 6.1% as medium, and 1.3% as high-risk, while MRPT-WS classified 92.9% as low, 6.4% as medium, and 0.7% as high-risk (*p* = 0.863). Among the 39 hospitalized patients that deteriorated, 30 were classified as high-risk or urgent by MPRT-WS, while 6 were classified as high-risk using NEWS [55].

Another observational study in two general wards of a large tertiary medical center included 217 high-risk patients for analysis [54]. Various scores were calculated, including NEWS, ABCNO (Airway, Breathing, Circulation, Neurology, Other), and a “wish list” deterioration score defined by the clinical team. Deterioration occurred in 24 patients: NEWS was high in 16 (67%), the ABCNO score was high in 18 (75%), and the “wish list” scoring criteria were observed in all 24 (100%) [54]. Of 193 patients without clinical deterioration, NEWS provided early warning alerts in 150 (77.7%), ABCNO criteria were met in 162 (83.9%), and the “wish list” criteria generated alerts for all 193 patients (100%) [54], corresponding to a sensitivity of 67% and specificity of 22% for NEWS; a sensitivity of 75% and specificity of 16% for ABCNO; and a 100% sensitivity and 0% specificity for the “wish list” criteria [54].

## 4. Discussion

AI supported CVSM have significant potential to improve patient care and reduce complications, but the clinical impact of AI supported CVSM on morbidity and mortality is sparsely described in the scientific literature. While all CVSM solutions claimed utilization of advanced AI, the assessed algorithms and methods mostly have potential application as simple AI, and only five studies from two CVSM solutions evaluated ML methods or prediction models with advanced AI potential, and these predictive algorithms do not appear to have been implemented in the solutions yet. Thus, a discrepancy between the claimed AI levels and the potential for the actual algorithms evaluated in peer-reviewed evidence was observed. This limitation is only relevant to current versions of the systems, with more advanced methods under development.

### 4.1. Vital Sign Thresholds and Algorithms for Prediction of Clinical Outcomes

Most studies used binary models of simple criteria for predictions of SAEs, clinical deterioration, and alarm filtering [14,15,43,48,55]. NEWS-based or other predefined thresholds demonstrated overall limited predictive ability for SAEs and generally resulted in many irrelevant alarms when applied to CVSM [14,15,39,46,51,54,55]. The lack of association between the duration and frequency of vital sign deviations outside predefined thresholds indicates that implementation of AI solely based on predefined thresholds is likely inadequate. It is still worth considering whether CVSM may outperform NEWS with this AI level, given that NEWS has only been designed and validated for predicting 24 h mortality, not for predicting SAEs, mortality outside 24 h, or further clinical deterioration [58]. Furthermore, CVSM significantly reduces the clinical staff’s workload from manual NEWS monitoring and is considered safe with minimal device-related adverse effects [59,60].

Eisenkraft 2023 used the Biobeat system and detected 25,635 readings (21.8% of the total) as “Urgent.” The clinical relevance of the utilized algorithms should be interpreted carefully, even considering health care providers did not respond to the alerts [55]. The study used healthy controls as a normal baseline for analysis and escalation scores, which may not be feasible since some degree of vital sign deviations are expected in all high-risk patients during admission [55]. Another study reported a significant imbalance in sensitivity and specificity for all evaluated algorithms (NEWS, ABCNO, “wish list”). The “wish list” algorithm generated alarms for all patients, yielding clinically irrelevant results with 100% sensitivity and 0% specificity [54].

Using the SV measuring RR, HR, and Tp, two studies analyzed cases of deteriorating patients included from observational studies and described vital sign patterns before specific AEs, defined as either preventable or unpreventable complications that required intervention [50,53]. The changes in all cases were not consistently detailed with thresholds but mostly limited to descriptions like “tachycardia” and “sudden RR increases”. Despite visualizations in figures that illustrate vital sign changes, these do not clarify thresholds or specific changes for complication prediction [50,53]. Describing and visualizing vital sign patterns in specific cases is likely insufficient to directly use as evidence for developing AI algorithms [17,26,27].

Two studies evaluated ML methods for the prediction of SAEs and clinical deterioration based on HR, RR, SpO_2_, Tp, and BP measurements from the WARD 24/7 system. Both studies found AUROC for the prediction of subsequent SAEs corresponding to a low accuracy, likely limiting the clinical relevance [34,35,61]. The limited predictive ability of these models was potentially due to heterogeneity in the SAEs or the fact that some SAEs may not at all be preceded by a vital sign deviation. The underlying pathophysiology varies substantially for different SAEs, even within specific categories. This variation likely results in different vital sign patterns preceding each specific SAE, making it impossible for one model to accurately predict all SAEs or a combination of various SAEs in specific groups.

### 4.2. Continuous Vital Sign Monitoring for Improving Clincial Outcomes

Posthuma 2024 conducted a large RCT using SV in high-risk surgical patients evaluating predefined thresholds, and found a 4.6% absolute reduction in new disability three months after surgery in the CVSM group, but only descriptive statistics were used, so differences between control and intervention groups should be interpreted with caution [48]. A feasibility RCT also utilized the SV system and reported the improvement of certain outcomes, but the small sample size limited the power to reach a significant conclusion [49].

Three before-and-after studies utilizing VM for the CVSM of HR, RR, SpO_2_, Tp, and BP, reported improved outcomes, including reduced ICU admissions, RRT calls, and complications [41,42,43]. The outcomes were not consistently improved across studies, and most other examined outcomes, including hospital length of stay and in-hospital mortality, were not significantly reduced [41,42,43]. Establishing a causal relationship between the intervention and outcomes is challenging with a before-and-after study design [62]. One study did not specify the method of patient monitoring during the baseline period [42], and another did not clarify the definition of complications and it may differ significantly from the AE and SAE definitions [41]. Another study utilized the VM system and conducted a propensity-matched retrospective observational analysis that demonstrated increased odds for adverse outcomes in the intermittently monitored group (mortality or ICU admission, heart failure, myocardial infarction, and acute kidney injury), but this study design was vulnerable to bias and still requires validation in large RCTs [11].

### 4.3. Filtering Alarms

Kjærgaard (2023) and Weller (2018) both reduced the number of alarms per patient per day through filtering and clinical staff feedback, respectively [39,46]. Neither study detailed the clinical impact of these alarm adjustments in terms of sensitivity and specificity, which is an issue since the goal should be to generate relevant alarms, not simply decrease the number of alarms. Another study utilized the VM system and created a cloud computing tool allowing threshold testing and optimization that may help manage alarm burden [40]. It remains uncertain if ML algorithms are applied to this database, and what the clinical impact would be of adjusting alarm thresholds [40].

Rossum (2022) utilized six different ML methods based on CVSM of RR, HR, and Tp with SV for adaptive threshold generation and successfully improved the sensitivity for any AE with a minimal increase in alarms, specifically when integrating multiple models as compared to the standard threshold based SV algorithm [38]. Although the study only included 39 patients, these findings are consistent with previous research indicating the potential for AI to reduce irrelevant alarms and predict complications [12,34,63].

### 4.4. Strengths and Limitations

The primary strength of this review is the assessment of many peer-reviewed articles providing a comprehensive overview of the evidence for different algorithms used within four different state-of-the-art CVSM solutions. We were unable to find other studies assessing current evidence with potential application as AI in CVSM, highlighting a gap between promised AI levels and the capabilities of current algorithms.

This review also comes with several limitations. The first limitation is the significant variety in study designs, CVSM sensors, outcomes, and reported performance metrics for the algorithms, making data comparison across studies and CVSM solutions difficult and rendering a meta-analysis unfeasible. The second limitation is the subjective aspects of AI definitions. No definitive guidelines exist on AI levels or the implementation of algorithms in peer-reviewed research to CVSM sensors. This leaves the AI levels claimed on official websites, and the potential AI level of the methods assessed in peer-reviewed evidence open to interpretation. A third limitation is the potential existence of more recent studies, as most included studies were found on the official CVSM solutions websites, and the latest literature may not have been uploaded or published yet. We screened and included many studies, with only two identified through the literature search not uploaded on the websites. The focus solely on non-invasive CVSM technologies and exclusion of specific patient groups, such as those in ICU and non-hospitalized settings, limits the generalizability of the study findings to the very high- and the low-risk populations. The results from this review will likely change over the coming months and years, where hopefully more CVSM solutions report results of the clinical validation of AI algorithms.

### 4.5. Perspectives

Several CVSM solutions, including those described in this review, have been implemented and validated in clinical settings, but there is a notable absence of large RCTs [64,65]. The assessment of CVSM for improving clinical outcomes needs to be thoroughly validated in large RCTs. There is currently a discrepancy in the definition of complications. A uniform method for defining complications, including AE and SAE, is warranted to make results transparent and comparable. As knowledge of AI expands and larger databases are developed, more sophisticated ML models will probably enhance the predictive ability of CVSM. Future benefits of AI will likely include stable clinical evaluations of both traditional and novel vital signs, such as heart rate variability and perfusion index, including personalized thresholds based on advanced data analyses of extensive databases [16,66,67], as well as integration with demographic data, blood samples, fluid therapy, and other clinically relevant factors [34,68].

All CVSM solutions evaluated in this review describe the application of medical evidence-based algorithms or AI to reduce alarms and improve patient outcomes [21,22,23,24,25]. This is currently only sparsely reflected in the number of peer-previewed publications, and the methods applied in the commercially available systems are still at a simple AI level. In addition, the experimental ML methods have generally presented a low accuracy for prediction of complications [34,35,61]. There are major limitations affecting how quickly AI can be applied to the commercial solutions, as both the conduction of large studies, analysis of data, and peer-review process take a long time and are associated with large financial expenses. The entire regulatory process of validating the effectiveness and safety of the algorithms is also very rigorous, complex, and time-consuming. Transparently disclosing detailed information about the AI and ML algorithms used to set alarm thresholds is important to ensure an understanding of assumptions and prevent AI biases from the data being unrepresentative to the applied population, which could lead to reduced effectiveness and potential adverse outcomes [69]. This further complicates their implementation, as the algorithms need to be validated in patient populations with different demographic characteristics to minimize AI bias [69].

## 5. Conclusions

Current evidence on CVSM primarily suggests its potential application as simple AI, which generates alarms based on fixed vital sign thresholds. All four CVSM solutions (100%) assessed algorithms with simple AI potential and only two CVSM solutions (50%) assessed hypothetical models with potential for application as advanced AI. A discrepancy exists between promised AI levels and the capabilities of algorithms assessed in the peer-reviewed evidence. Transparent reporting on the application of AI and ML algorithms in CVSM sensors is warranted.

## Figures and Tables

**Figure 1 sensors-24-06497-f001:**
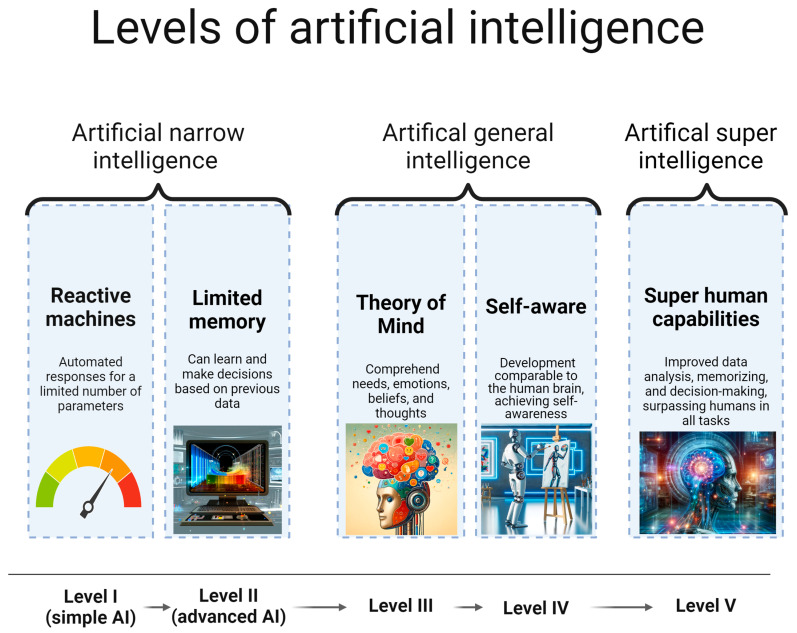
Levels of artificial intelligence. Legends: Created in BioRender. Aagaard, N. (2024) BioRender.com/q01h863. Based on standardized AI definitions [26]. Pictures representing level II–V were created using DALL·E 3 (OpenAI, San Francisco, CA, USA).

**Figure 2 sensors-24-06497-f002:**
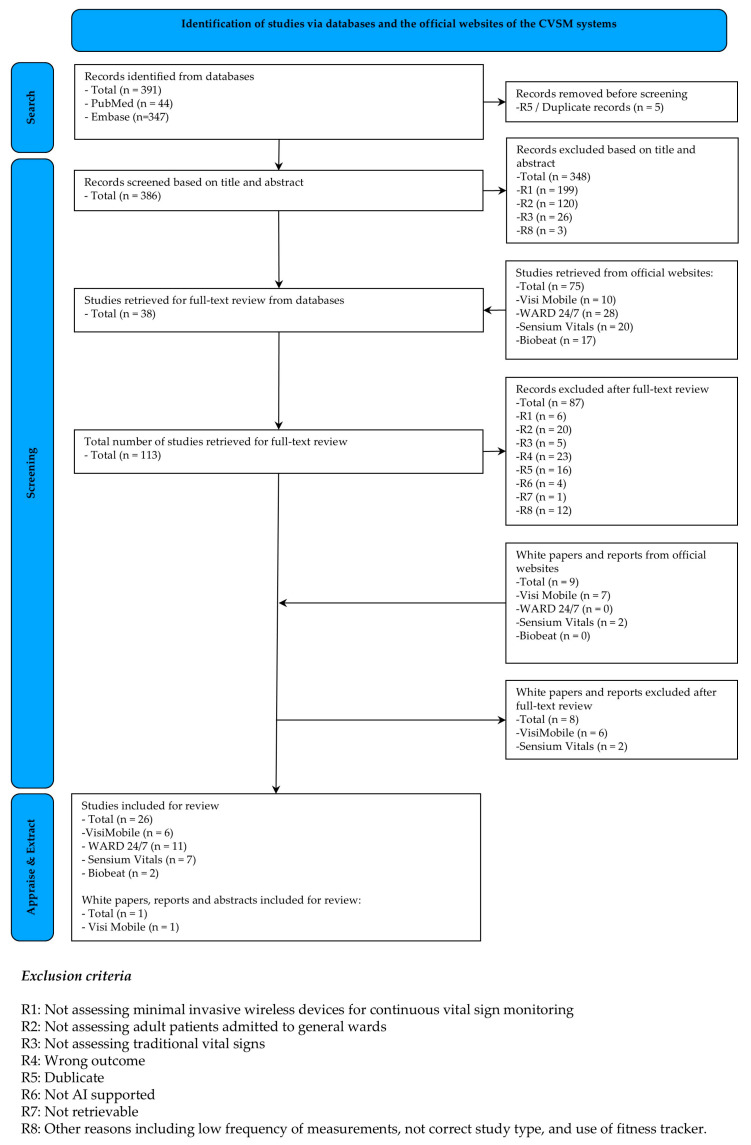
Prisma flow diagram showing the selection process of included articles.

**Figure 3 sensors-24-06497-f003:**
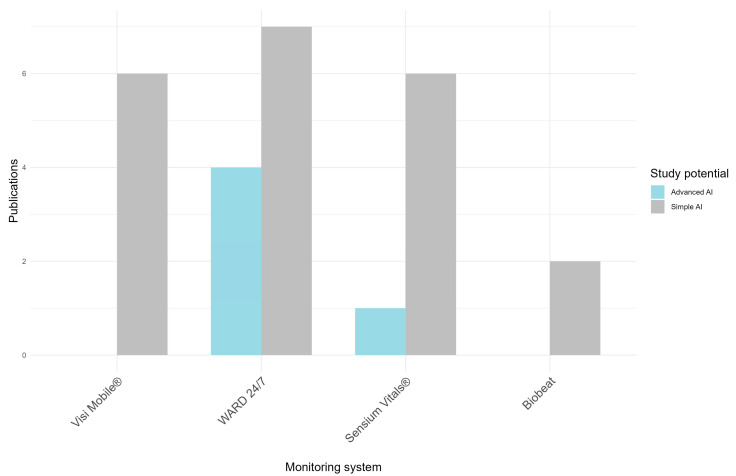
Number of peer-reviewed studies assessing algorithms for potential simple and advanced AI applications.

**Table 1 sensors-24-06497-t001:** Search report.

Search Report	Current Evidence on Continuous Vital Sign Monitoring of In-Hospital Patients
Database	PubMed and Embase
Date conducted	30 May 2024
Search terms in PubMed	Continuous * monitor * OR “physiologic monitoring” OR “physiological monitoring” OR real-time monitoring OR automat * monitor * OR “surveillance monitoring” OR “signal processing” AND (randomized controlled trial [Publication Type] OR observational study [Publication Type] OR prospective observational study [Publication Type] OR cluster randomized controlled trial [Publication Type]) AND (Blood Pressure [MeSH Terms] OR Oxygen Saturation [MeSH Terms] OR Body Temperature [MeSH Terms] OR Heart Rate [MeSH Terms] OR Respiratory Rate [MeSH Terms] OR Vital Signs [MeSH Terms]) AND wireless
Search terms in Embase	Continous vital sign monitoring {Including Related Terms}
Limits in Embase	Limit to (human and english language and embase and (clinical trial or randomized controlled trial or controlled clinical trial or multicenter study) and yr = “2010-Current” and article and (adult <18 to 64 years> or aged <65+ years>))
Hits	391
Included	10

**Table 2 sensors-24-06497-t002:** Characteristics of continuous vital sign monitoring systems.

Company	Devices	Country	CE-Mark	FDA-Approved or Cleared	Population in Published Articles		Study Types in Peer-Reviewed Evidence			Vital Signs Measured				
					Surgical	Medical	RCT	OBS	B-A	RR	SpO2	HR	BP	TP
Sotera Digital Health	Visi Mobile^®^	Carlsbad, California, United States	x	x	x	x	x	x	x	x	x	x	x	x
WARD 247 ApS	Isansys Lifetouch and Lifetemp, Nonin WristOx, 3150 Meditech BlueBP-05, or A&D TM-2441	Copenhagen, Denmark	x		x	x		x		x	x	x	x	x
The Surgical Company, Connected Care	Sensium Vitals^®^	Abingdon, United Kingdom	x	x	x	x	x	x		x		x		x
Biobeat Technologies Ltd.	Biobeat wrist monitor and chest monitor	Petah Tikva, Israel	x	x	x	x		x		x	x	x	x	x

Legends: FDA: Food and Drug Administration, RCT: randomized controlled trial, OBS: observational studies, B-A: before-and-after studies, RR: respiratory rate, SpO_2_: peripheral oxygen saturation, HR: heart rate, BP: blood pressure, TP: temperature (skin or axillary).

**Table 3 sensors-24-06497-t003:** Comparison of artificial intelligence levels stated on official websites versus levels found in peer-reviewed evidence.

System and Source	Citation of AI Level on Official Website	Stated AI Level on Official Website	Comparison of AI-Levels Stated on Websites versus Published in Peer-Reviewed Journals	Potential AI Level in Peer-Reviewed Evidence	Description of Methods Utilized in Peer-Reviewed Evidence	Number of Studies Using Algorithms with Potential for Promised AI Level
Visi Mobile^®^	*“ViSi Mobile improves patient safety by utilizing machine learning from millions of hours of patient data to recognize patient deterioration while minimizing alarm burden and maximizing clinical workflow”*	Advanced AI (Level II)	>	Simple AI (Level I)	Studies using predefined thresholds with adjustment based on clinical staff feedback to reduce total alarms.	0
WARD24/7.org	*“Wireless Assessment of Respiratory and circulatory Distress (WARD) offers continuous 24/7 observation of high-risk patients with real-time alarms through innovative machine learning algorithms in a Clinical Support System (CSS)”*	Advanced AI (Level II)	=	Advanced AI (Level II)	Advanced ML algorithms to predict deterioration and SAEs. Not clearly stated if implemented to the clinical support system yet.	4
WARD24/7.com	*“Advanced clinically modelled algorithms interpret standard vital sign patterns from standard wireless sensors and detect deterioration earlier in real-time”*	Advanced AI (Level II)	=	Advanced AI (Level II)	Advanced ML algorithms to predict deterioration and SAEs. Not clearly stated if implemented to the clinical support system yet.	4
Sensium Vitals^®^	*“Sensium’s monitoring and smart algorithms continuously process and analyse all patient data, generating targeted notifications of deterioration, efficiently bringing the nurse to the deteriorating patient.”*	Advanced AI (Level II)	=	Advanced AI (Level II)	Adaptive thresholds-based alarm strategies, and a combination of these strategies. Not clearly stated if implemented to the system yet.	1
Biobeat	*“Biobeat’s solution uses health-AI and ML on big-data in order to provide actionable insights on patient care. More than just analyzing the data, Biobeat also generates it, using our proprietary sensor for continuous monitoring of vital signs unique to Biobeat”*	Advanced AI (Level II)	>	Simple AI (Level I)	Studies using predefined thresholds based on a modification of NEWS or fixed thresholds retrospectively set by a specialist panel	0

Legends: Equal and greater than signs illustrate comparison of AI-levels stated on websites and published in peer-reviewed journals. ML: machine learning; SAEs: serious adverse events; ECG: electrocardiographic; AI: artificial intelligence; N.A.: not available.

**Table 4 sensors-24-06497-t004:** Overview of alarms based on system and study.

Alarm	Thresholds	Duration before Alarm
**Visi Mobile [39]**		
Bradypnea	RR < 4 breaths/min	2 min
Tachypnea	RR > 35 breaths/min	2 min
Desaturation	SpO_2_ < 85%	1.5 min
Tachycardia	HR > 150 beats/min	15 s
Bradycardia	HR < 39 beats/min	15 s
Hypotension	MAP < 58 mmHg	1 min
Hypertension	SYS BP > 200 mmHg	4 min
**WARD 24/7 [46]**		
Bradypnea	RR ≤ 5 breaths/min and HR ≥ 20 beats/min	1 min
Tachypnea	RR ≥ 24 breaths/min	5 min
Desaturation	SpO_2_ < 92%	60 min
	SpO_2_ < 88%	10 min
	SpO_2_ < 85%	5 min
	SpO_2_ < 80%	1 min
Sinus tachycardia	HR ≥ 111 beats/min	60 min
	HR > 130 beats/min	30 min
Bradycardia	HR < 30 beats/min	1 min
	HR = 30–40 beats/min	5 min
Hypotension	SYS BP <70 mmHg	1 min
	SYS BP <91 mmHg	30 min
Hypertension	SYS BP ≥180 mmHg	60 min
	SYS BP ≥220 mmHg	1 min
**Sensium Vitals [38]**		
Alarm strategy	Specification	Parameter setting
Threshold individualization	Alarm thresholds were defined using the cumulative density function (CDF), for each vital sign based on the first 24-h monitoring. Lower and upper alarm vital sign thresholds corresponds to the lower (*CDF_low_*) and upper (*CDF_high_*) percentiles of the individual CDF. Default alarm thresholds were used for the first 24 h.	*- CDF_low_:* 0.1%; *CDF_high_:* 99.9% *- CDF_low_:* 0.5%; *CDF_high_:* 99.5% *- CDF_low_:* 1%; *CDF_high_:* 99%
Postoperative elevation of upper thresholds	The standard upper alarm threshold is increased by a fixed percentage (*PO_increase_)* i.e., postoperative increase factor) for the first four days after surgery.	*- PO_increase_:* 5% for HR/RR; 1% for T *- PO_increase_:* 10% for HR/RR; 2.5% for T *- PO_increase_:* 25% for HR/RR; 5% for T
Increase annunciation delay interval	The length of the annunciation delay interval (*L_interval_*) i.e., minimum number of successive abnormal measurements needed for generation of an alarm (default: 7 measurements, i.e., 14 min interval) is increased.	*- L_interval_:* 12 measurements *- L_interval_:* 17 measurements *- L_interval_:* 22 measurements
Daytime elevation of upper HR/RR thresholds	The standard upper HR and RR threshold is increased by a fixed percentage (*DT_increase_*)i.e., daytime increase factor) during daytime (8 a.m. to 10 p.m.).	*- DT_increase_*: 5% for HR; 15% for RR *- DT_increase_:* 10% for HR; 25% for RR *- DT_increase_:* 25% for HR; 35% for RR
Nighttime reduction of lower HR/RR thresholds	The standard lower HR and RR threshold is decreased by a fixed percentage (*NT_increase_*) i.e., nighttime decrease factor) during nighttime (10 p.m. to 8 a.m.).	*- NT_decrease_*: 5% for HR; 15% for RR *- NT_decrease_*: 10% for HR; 25% for RR *- NT_decrease_:* 25% for HR; 35% for RR
Slope-based alarms	An alarm is generated only in case the slope of the linear regression line calculated over a past time interval (*T_slope_*) exceeds a preset threshold: HR slope: ± 15 bpm over *T_slope_*; RR slope: ± 10 brpm over *T_slope_*; T slope: ± 1 °C over *T_slope_.*	*- T_slope_:* 4 H *- T_slope_:* 8 H *- T_slope_:* 12 H

Legends: RR: respiratory rate; SpO_2_: peripheral oxygen saturation; HR: heart rate; BP: blood pressure; TP: temperature; SYS: systolic; Min: minutes; Sec: seconds; H: hours; mmHg: millimeter of mercury.

## Data Availability

No new data were created or analyzed in this study. Data sharing is not applicable to this article.

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
