# Peer review of "Discrepancies between Promised and Actual AI Capabilities in the Continuous Vital Sign Monitoring of In-Hospital Patients: A Review of the Current Evidence"

_sensors, 2024, doi:10.3390/s24196497_

Round 1
Reviewer 1 Report
Comments and Suggestions for Authors
The paper give a comprehensive review on application of artificial intelligence in continuous vital sign monitoring of in-hospital patients. but the literature search strategy, inclusion/exclusion criteria, and classification of AI levels should be more thoroughly explained. Additionally, a more detailed description of the study selection, data extraction processes, and the assessment of AI algorithms is necessary. Including a formal quality assessment of the included studies, as well as addressing how non-peer-reviewed sources were evaluated, would improve the credibility of the findings. Strengthening these areas will make the methodology more robust and allow for greater confidence in the results.
1 The description of the search strategy is brief and could benefit from additional details. For example, in the section on database searches, it would be helpful to specify the time frame used for the literature search. Were studies from certain years prioritized or excluded? How were the search terms chosen, and were they tested for efficacy before the final search was conducted?
2 The inclusion and exclusion criteria are not fully explained. More specific criteria for the selection of studies would be useful, particularly regarding the population (e.g., why limit to general wards and exclude ICU or high-dependency units?) and the intervention (e.g., non-invasive biosensors). Explain why some patient groups (e.g., outpatients, ICU patients) were excluded and how this decision impacts the generalizability of your findings.
3 Since the results were presented descriptively due to the heterogeneity of the studies, it would be beneficial to explain why a meta-analysis was not feasible. Discuss whether any efforts were made to standardize the data for a potential quantitative synthesis (e.g., via subgroup analyses or pooling similar outcome measures), and if not, clarify why this decision was made.
4 Fluid detection is also an important part of patient monitoring, such as patient blood flow rate, blood bubble monitoring, and research in this area also needs to be cited, https://doi.org/10.1109/TIM.2024.3372226.
Reviewer 2 Report
Comments and Suggestions for Authors
This study is a type of meta-analysis that examines the clinical efficacy of artificial intelligence algorithms integrated with Continuous Vital Sign Monitoring (CVSM) based on previously published data. Understanding the current status of the effectiveness of AI algorithms in real clinical environments, which might be overlooked by engineers focused on sensor-related fundamental or applied technologies, is a meaningful effort. However, this paper, in its current form, does not contain sufficient information to be published in the Sensor journal.
Most importantly, the conclusion of this study reports a type of negative result.
Specifically, the authors have found that it is challenging to find cases where AI algorithms are properly applied to Continuous Vital Sign Monitoring (CVSM) based on the literature analyzed. Moreover, even when such algorithms are applied, there is a lack of clear reporting on the content and performance of these AI algorithms. In fact, the "simple AI" defined by the authors can be considered as conventional signal processing techniques. For "advanced AI," which can be considered a genuine AI algorithm, only 50% (2 products) of the CVSMs analyzed met this criterion, which is insufficient to draw meaningful conclusions. The final conclusion, which calls for transparent reporting on the application of AI algorithms to CVSMs, is derived from reports on commercialized systems and may not provide meaningful information to most readers of the journal.
Comments on the Quality of English LanguageThe general English is written at a level that is sufficiently understandable, but there are some typos( such as line 37 : implented, line 58 : decission, line 102 : og ), so corrections and improvements are needed.
Round 2
Reviewer 1 Report
Comments and Suggestions for Authors
The manuscript has been well revised
Author Response
The manuscript has been well revised.
Response: Thank you once again for the constructive feedback. We are pleased to hear that our revisions have adequately addressed your comments.
Reviewer 2 Report
Comments and Suggestions for Authors
Even after the first revision, the core content of the paper did not change.
As the authors pointed out, in the absence of large-scale RCTs, it is basically impossible to investigate whether AI-supported CVSM has significant potential to improve patient care and reduce complications through comparison with conventional CVSM (or manual monitoring). So they can only end up pointing out discrepancies between website reports and the evidence that reached publications as peer-reviewed evidence.
The paper title needs to be replaced with a more accurate one reflecting the fact that the article reviews the current evidence and highlights the discrepancy between the promises and actual performance of AI in CVSM, rather than proving its clinical superiority. For example, "Discrepancies Between Promised and Actual AI Capabilities in Continuous Vital Sign Monitoring: A Review of Published Evidence"
Author Response
Even after the first revision, the core content of the paper did not change.
As the authors pointed out, in the absence of large-scale RCTs, it is basically impossible to investigate whether AI-supported CVSM has significant potential to improve patient care and reduce complications through comparison with conventional CVSM (or manual monitoring). So they can only end up pointing out discrepancies between website reports and the evidence that reached publications as peer-reviewed evidence.
The paper title needs to be replaced with a more accurate one reflecting the fact that the article reviews the current evidence and highlights the discrepancy between the promises and actual performance of AI in CVSM, rather than proving its clinical superiority. For example, "Discrepancies Between Promised and Actual AI Capabilities in Continuous Vital Sign Monitoring: A Review of Published Evidence".
Response: Thank you for the thorough review. We have revised the title to reflect the findings and focus of the paper (page 1, line 2-4): ”Discrepancies Between Promised and Actual AI Capabilities in Continuous Vital Sign Monitoring of In-Hospital Patients: A Review of Current Evidence”.
It is correct that the evidence presented in this manuscript cannot prove whether AI-supported CVSM has benefits compared with conventional CVSM or manual monitoring. Therefore, we have modified the aim both in the introduction and the abstract. Please see the abstract (page 1, line 15-16): “With the aim of summarizing peer-reviewed evidence for AI-support to CVSM sensors” and the introduction (page 1, line 61-63): “This review aims to summarize current type of evidence on CVSM sensors in hospitalized patients with potential for application as AI algorithms within state-of-the-art CVSM solutions”. We hope this further underlines that the manuscript investigates discrepancies between promised and actual AI capabilities in CVSM.